# Wound-Dressing-Based Antenna Inkjet-Printed Using Nanosilver Ink for Wireless Medical Monitoring

**DOI:** 10.3390/mi13091510

**Published:** 2022-09-12

**Authors:** Chun-Bing Chen, Hsuan-Ling Kao, Li-Chun Chang, Yi-Chen Lin, Yung-Yu Chen, Wen-Hung Chung, Hsien-Chin Chiu

**Affiliations:** 1Department of Dermatology, Drug Hypersensitivity Clinical and Research Center, Chang Gung Memorial Hospital, Linkou, Taipei and Keelung Branches, Taoyuan City 33305, Taiwan; 2Cancer Vaccine and Immune Cell Therapy Core Laboratory, Chang Gung Memorial Hospital, Linkou, Taoyuan City 33305, Taiwan; 3Chang Gung Immunology Consortium, Chang Gung Memorial Hospital and Chang Gung University, Taoyuan City 33305, Taiwan; 4Department of Dermatology, Xiamen Chang Gung Hospital, Xiamen 100084, China; 5College of Medicine, Chang Gung University, Taoyuan City 33302, Taiwan; 6Whole-Genome Research Core Laboratory of Human Diseases, Chang Gung Memorial Hospital, Keelung 204, Taiwan; 7Immune-Oncology Center of Excellence, Chang Gung Memorial Hospital, Linkou, Taoyuan City 33305, Taiwan; 8Genomic Medicine Core Laboratory, Chang Gung Memorial Hospital, Linkou, Taoyuan City 33305, Taiwan; 9Graduate Institute of Clinical Medical Sciences, College of Medicine, Chang Gung University, Taoyuan City 33302, Taiwan; 10School of Medicine, National Tsing Hua University, Hsinchu 300044, Taiwan; 11Department of Dermatology, Chang Gung Memorial Hospital, Linkou Branch, Taoyuan City 33305, Taiwan; 12Department of Electronics Engineering, Chang Gung University, Taoyuan City 33302, Taiwan; 13Centre for Reliability Sciences and Technologies, Chang Gung University, Taoyuan City 33302, Taiwan; 14Department of Materials Engineering, Ming Chi University of Technology, New Taipei 243303, Taiwan; 15Center for Thin Film Technologies and Applications, Ming Chi University of Technology, New Taipei 243303, Taiwan; 16Department of Electronic Engineering, Lunghwa University of Science and Technology, Taoyuan City 333326, Taiwan; 17Department of Dermatology, Beijing Tsinghua Chang Gung Hospital, School of Clinical Medicine, Tsinghua University, Beijing 102218, China; 18Department of Dermatology, Ruijin Hospital, School of Medicine, Shanghai Jiao Tong University, Shanghai 200025, China

**Keywords:** inkjet printing, silver film, antenna, wound dressing

## Abstract

In this paper, we present a wound-dressing-based antenna fabricated via screen-printed and inkjet-printed technologies. To inkjet print a conductive film on wound dressing, it must be screen-printed, UV-curable-pasted, and hard-baked to provide appropriate surface wettability. Two passes were UV-curable-pasted and hard-baked at 100 °C for 2 h on the wound dressing to obtain 65° WCA for silver printing. The silver film was printed onto the wound dressing at room-tempature with 23 μm droplet spacing for three passes, then sintered at 120 °C for 1 h. By optimizing the inkjet printing conditions by modifying the surface morphologies and electrical properties, three-pass printed silver films with 3.15 μm thickness and 1.05 × 10^7^ S/m conductivity were obtained. The insertion losses at the resonant frequency (17 and 8.85 GHz) were −2.9 and −2.1 dB for the 5000 and 10,000 μm microstrip transmission lines, respectively. The material properties of wound dressing with the relative permittivity and loss-tangent of 3.15–3.25 and 0.04–0.05, respectively, were determined by two transmission line methods and used for antenna design. A quasi-Yagi antenna was designed and implemented on the wound-dressing with an antenna bandwidth of 3.2–4.6 GHz, maximal gain of 0.67 dBi, and 42% radiation efficiency. The bending effects parallel and perpendicular to the dipole direction of three fixtures were also examined. The gain decreased from 0.67 to −1.22 dBi and −0.44 dBi for a flat to curvature radius of 5 cm fixture after parallel and perpendicular bending, respectively. Although the maximal gain was reduced with the bending radius, the directivity of the radiation pattern remained unchanged. The feasibility of a wound-dressing antenna demonstrates that inkjet-printed technology enables fast fabrication with low cost and environmental friendliness. Additionally, inkjet-printed technology can be combined with sensing technology to realize remote medical monitoring, such as with smart bandages, for assessment of chronic wound status or basic physical conditions.

## 1. Introduction

The rapid development of the Internet of Medical Things (IoMT) enables medical services to be conducted remotely and monitored over long periods of time. Wearable electronic devices for monitoring of various biological signals are rapidly gaining popularity. Additionally, wireless transmission provides effective remote medical monitoring, simplifying complex transmission systems. Integrating wireless transmission and wearable medical devices provides the IoMT with versatile wireless technology and energy-saving features, making it the foundation for next-generation healthcare [1,2].

Wearable medical monitoring devices must be lightweight and small in size, with optimal flexibility, breathability, and disposability to ensure comfort for skin, which is dependent on the choice of substrate. Flexible antennae represnet a key technology in remote healthcare monitoring, providing continuous wireless tracking and transmission [3]. Fabrication of flexible antennae by stitching, weaving, embroidery, screen printing, and inkjet printing has been reported, related to the substrate material. Dual-band monopole patch stitching has been proposed using copper sheets on 1 mm jean fabric. An EBG substrate was used as a barrier to reduce 15 dB radiation in the human body [4]. A total of 160 mm diameter Archimedean spiral antennae and a ground plane were woven by seven-filament silver-plated copper Elektrisola e-threads on a 0.59 mm thick Kevlar fabric substrate with a permittivity of 2.6 and a loss tangent of 0.006, and urethane foam was placed between the spiral and ground to provide a reflecting cavity. A polarized gain of 6.5 dBi and a bandwidth of 0.3–3 GHz were obtained [5]. A passive UHF RFID tag was realized by stitching e-textile and connecting the antenna to the IC interfaces by a conductive thread. Stable electromagnetic performance was observed in different body-worn configurations [6]. A 5 GHz 20% BW textile-integrated waveguide with a slotted, coaxial-fed antenna fabricated by mixed embroidery–weaving methods has also been reported. Polyethyleneterephthalate and polyethersulfone yarns have been used as dielectric materials, and two kinds of Shieldex 117f17 yarns were used as the conductive layer. The size of the textile integrated waveguide was 28 × 30 × 0.36 mm^2^ [7].

However, the processing of fabric antennae by embroidery or weaving is complicated and difficult to achieve in terms of size minimization and accuracy; thus, such antennae are typically used below the 2.4 GHz bandwidth. Printing processes such as screen printing and inkjet printing can simplify the production of flexible antennae. Screen printing is a simple, fast, and low-cost technique to implement antennae on various flexible substrates. Rectennae were fabricated on a 350 μm polycotton fabric substrate. The conductive layer was screen-printed with thermally curable silver polymer ink onto a UV-curable interface layer, and a radiation efficiency of 11% was achieved at 2.45 GHz [8]. A graphene-flake, quasi-dipole antenna was screen-printed onto PI substrates, with a sheet resistance of graphene of 4 Ω/sq at 10 μm thickness after annealing at 350 °C for 30 min. The graphene-flake antenna was operated at 2–5 GHz with a maximum gain of 2.3 dB [9]. A parasitic antenna array with PRS on a low-loss flexible material demonstrating a high gain of 11.2 dBi at 77 GHz was realized by screen-printing technology. The single patch length and width were 1.13 mm, with a feed line width of 0.45 mm [10].

Inkjet printing technology is not only an additive process but also enables the direct printing of digitally patterned materials without a mask. This technology can be used to print various materials on diverse substrates, including deformable substrates, providing a fast, environmentally friendly, and cost effective production [11,12,13]. Several reports have presented studies on inkjet-printed flexible antennae. A 60 GHz coplanar square monopole antenna was inkjet-printed on PEN with 68% efficiency and 1.8 dBi gain [14]. An inkjet-printed patch antenna was proposed with an MWNT/PEDOT:PSS matching line on a PDMS substrate operating at 13 GHz [15]. 3D multidirector Yagi–Uda antennae were inkjet-printed on a flexible liquid crystalline polymer with an 8 dBi gain with silver conductive and SU-8 dielectric inks, alternately, demonstrating vertically integrated antenna structures for high-frequency wireless electronics [16]. A coplanar-waveguide-fed, Z-shaped planar antenna was fabricated on PET with a gain of 1.44 dBi at 2.45 GHz and a radiation efficiency of 62% using inkjet printing technology for IoT applications [17]. According to previous reports, the performance of flexible antennae is primarily related to substrate properties, film material, and fabrication techniques.

To provide comfort for long-term wireless medical monitoring, hydrocolloid semi-permeable wound dressing was used in the present study. Wound dressings generally offer enhanced flexibility and breathability and are cheap and easy to obtain. Wound dressing can be combined with sensing technology to continuously detect status information, such as that related to chronic wounds or blood pressure, through wireless transmission, such as smart bandages, providing features that are simple to manufacture, lightweight, flexible, and disposable. However, the surface of wound dressing is rough and semi-permeable, which is not conducive to inkjet printing. To overcome these limitations, an interfacial layer, such as a UV-curable paste [18,19,20], is screen-printed before inkjet printing. Planar quasi-Yagi antennae with a partial ground conductor at the bottom provide a wide bandwidth, high gain, low cost, and ease of manufacturing. In this paper, we present an inkjet-printed, wound-dressing-based quasi-Yagi antenna. Two transmission zeros were produced by filtering balun. The characteristics of the wound-dressing-based antenna, including bending effect, were also examined. Table 1 summarizes the performance of the proposed wound-dressing-based antenna and that of other reported designs [4,5,6,7,8,9,10,14,15,16,17] for comparison. Inkjet-printed wound-dressing antennae offer comfort and practicality for wearable wireless medical monitoring devices.

This paper is structured as follows. In Section 2, we introduce the materials, fabrication process, and characterization methods. In Section 3.1, we describe the surface morphologies, cross section, and the water contact angle of the screen-printed, UV-curable paste. In Section 3.2, we presents the surface morphologies, cross section, and electrical properties of the inkjet-printed silver film with various droplet spacings and passes. In Section 3.3, we focus on the applications of the proposed inkjet-printed, wound-dressing-based antenna, including the bending effect. Conclusions are presented in Section 4.

## 2. Experimental Section

### 2.1. Materials

Tegaderm^TM^ hydrocolloid thin wound dressing, a hydrocolloid, moisture-retentive wound dressing, was used in this study. Figure 1 shows an optical micrograph and the surface morphology of hydrocolloid thin wound dressing; a rough surface was observed for the wound dressing. Figure 2 shows silver film inkjet-printed onto the wound-dressing. The inkjet-printed silver film is discontinuous because of the rough surface. A part of the silver film even immerses into the wound dressing due to its semi-permeability. To avoid immersion and improve the surface roughness, a polyurethane-based, ultraviolet-curable paste (Fabink UV-IF1, Smart Fabric Inks Ltd., Southampton, UK) was used as an interface layer via screen-printing technology to provide an appropriate surface for subsequent inkjet printing. Silver nanoparticle ink (DGP-40LT-15C) containing 30–35% silver solids dissolved in triethylene glycol monoethyl ether with a viscosity of 10–17 cPs and a surface tension of 35–38 dyne/cm was purchased from ANP for inkjet printing of the conductive films using an inkjet printer. A 16-jet cartridge (DMC-11610) with a diameter of 20 µm was used for each nozzle to generate 10 pL drops of ink to obtain high-resolution conductive patterns.

### 2.2. Fabrication of Inkjet-Printed Silver Films

Figure 3 shows the process flow of inkjet printing of silver films onto a hydrocolloid thin wound dressing. Screen-printing technology was used for the UV-curable paste. The UV-curable paste was placed by a dropper and spread by a squeezing plastic pipette onto the wound dressing, followed by UV exposure for 30 min under uniform pressure and a subsequent 30 min of hard baking at 150 °C for curing. Uniform pressure was applied using two glasses. Second, screen-printing was repeated to obtain a uniform surface of UV-curable paste. To exhibit appropriate surface wettability of the UV-curable paste, the hard-baked paste was treated in an oven to change its surface wettability for subsequent inkjet printing. Then, the silver nanoparticle ink was filtered into a cartridge and inkjet-printed on the heated substrate to obtain a uniform silver film. Multipass printing of the silver film was required to improve the uniformity and conductivity of the silver film. Finally, the silver films were sintered at 120 °C for 1 h to improve their conductivity.

### 2.3. Characterization of UV-Curable Paste and Silver Film

Water contact angle (WCA) measurements were used to characterize the surface wettability of the UV-curable paste after hard baking. A scanning electron microscopy (SEM, Hitachi S-3400N, Nara, Japan) with a 15 kV accelerating voltage was used to examined the surface morphologies and cross sections of the UV-curable paste and silver film. Samples were prepared by cold mounting process for a cross-section view. A four-probe method was used to measuring the sheet resistance of the silver film. High-frequency measurement was conducted with an HP 8510C network analyzer were for silver lines to calculate the relative permittivity and loss tangent of the hydrocolloid thin wound-dressing with UV-curable paste. An antenna on the wound dressing was characterized in terms of its radiation patterns inside a far-field anechoic chamber. The bending effect of the antenna was also examined by using bending fixtures with curvature radii of r = 5 and 6 cm.

## 3. Results

### 3.1. Screen Printing of UV-Curable Paste

Due to the rough surface and hydrophilic nature of wound dressing, a UV-curable paste was fabricated onto the dressing as an interfacial layer via screen-printing technology. Figure 4 shows the surface morphologies of one- and two-pass UV-curable pastes on wound-dressing. A slight roughness was observed for the one-pass UV-curable paste, and a uniform surface was achieved after two passes. Figure 5 shows a cross-sectional view of the UV-curable paste after one and two passes. The thicknesses of the UV-curable pastes after one and two passes were 163–173 and 252–270 μm, respectively. The wound-dressing thickness was about 444–498 μm. However, the initial WCA of the UV-curable paste after screen printing was 56.78°. To provide the appropriate wettability for the subsequent inkjet printing, hard baking was used to change the surface from hydrophilic to slightly hydrophilic. Figure 6 shows WCA versus temperature for 0.5 and 2 h of hard baking on the UV-curable paste. The WCA increased with temperature and time. To provide a uniform silver film of sufficient thickness, 65° WCA was selected for subsequent experiments.

### 3.2. Inkjet-Printed Silver Film

The parameters, including droplet spacing and substrate temperature, should be optimal to provide a high-quality inkjet-printed film. Inkjet liquid droplets should overlap to coalesce a continuous film by controlling the droplet spacing. Figure 7 shows optical photographs of the silver film with defined pattern line widths of 70 and 400 μm for one-pass printing with droplet spacing of 10 to 30 μm. A bulge was observed in association with smaller droplet spacing (<20 μm) because the excessive aggregation of droplets led to a collapse, making it difficult to define the width. The edges of the silver film were jagged with droplet spacing of more than 25 μm, as wide droplet spacing made the droplets unable to connect; thus, the film could not be continuous. A void was also observed with a droplet spacing of 30 μm and a pattern width of 400 μm. The measured width was approximately a 400 μm with a droplet spacing of 20–23 μm. However, the measured width with droplet spacing of 23 μm was 102 μm, which is closer to the pattern width of 70 μm than the silver film with a droplet spacing of 20 μm. A slight bulge was observed on the silver film with a droplet spacing of 20 μm and a pattern width of 70 μm. Based on the optical photographs, a 23 μm droplet spacing was used to obtain a continuous and uniform silver film.

The thickness and conductivity of the silver film obtained with one-pass printing were insufficient for printing with nanosilver ink. Stacking the silver film by multipass printing increased the film thickness and improved the surface roughness to obtain better electrical properties. However, wound dressing is soft and can therefore sink after the the silver film is printed. Silver film is difficult to reflow on a sunk surface, resulting in a rough silver surface with holes. To improve the surface of silver film, it is necessary to wait for approximately 3 h for the wound dressing to recover to room temperature after silver film printing, followed by hard baking at 45 °C for silver ink reflow to provide a uniform film surface and improve conductivity. Figure 8 shows the surface morphologies of silver films obtained with one to five passes. Obvious holes were observed after one pass, as the silver film was thin and dried within 3 h. However, the number of holes reduced after two passes, and a uniform surface was obtained after three passes. After three passes, a slight deterioration in surface uniformity was observed, although not affecting the conductivity, as reflow at 45 °C is insufficient to achieve a thicker silver film. Therefore, three passes with 45 °C baking represent the optimal conditions for subsequent fabrication. Figure 9 shows a cross-sectional view of silver films obtained with one to five passes; thickness increases proportionally with the number of passes. Figure 10 shows the sheet resistance and thickness of silver films obtained with one to five passes. The sheet resistance decreased sharply for one to three passes, as the surface roughness was improved by the multipass method, which is consistent with the observed surface morphologies. The smoothness decreased proportional to the thickness with three to five passes. Furthermore, a slight deterioration in the surface uniformity following four and five passes did not affect the sheet resistance. Based on the sheet resistance and thickness, the conductivity of the silver film was calculated as 1.05 × 10^7^ S/m for three passes.

Two microstrip transmission lines the lengths of 5000 and 10,000 μm, respectively, and a width of 100 μm were fabricated on the wound dressing by screen-printing and inkjet-printing technologies, respectively. The S parameters of the microstrip line were measured up to 50 GHz using an HP 8510C network analyzer. The relative permittivity and loss tangent were calculated by phase difference and the ratio of S_21_ of the long and short lines, respectively. The calculated relative permittivity and loss tangent were 3.15–3.25 and 0.04–0.05, respectively, according to the two-transmission-line method [21,22,23]. Figure 11 shows the measured and simulated insertion and return loss of two silver lines on a wound dressing. By substituting the calculated relative permittivity and loss tangent into the software (Ansoft High Frequency Structure Simulator, HFSS), the measured and simulated results were found to be similar, thereby proving the accuracy of the extracted parameters. The resonant frequencies were 8.85 and 17 GHz for the 5000 and 10,000 μm lines, respectively. The insertion losses at the resonant frequency were −2.9 and −2.1 dB for the 5000 and 10,000 μm lines, respectively. Based on the two-transmission-line method, the relative permittivity and the loss tangent of a hydrocolloid thin wound dressing with a UV-curable paste can be calculated by the phase difference and insertion loss ratio.

### 3.3. Inkjet-Printed Wound-Dressing-Based Antenna

Quasi-Yagi antennae have been used extensively as end-fire antennae, which provide increased performance over dipole antennae when a high concentration of radiation is desired in a certain direction. In addition, quasi-Yagi antennae have a simple structure and low profile and are easy to fabricate and lightweight, making them suitable for disposable medical applications. To provide a wireless transmission for medical applications, an inkjet-printed quasi-Yagi antenna was implemented on wound dressing. Figure 12 shows the geometrical configuration of the proposed quasi-Yagi antenna, consisting of a feed line, balun, coplanar strip line, dipole, and director. A 3D electromagnetic simulation was generated by an HFSS. The designed antenna was simulated on the wound dressing with a UV-curable paste in free space using the parameters of ε_r_= 3.2, tan δ= 0.05, substrate thickness = 700 μm, and conductor conductivity = 1.05 × 10^7^ S/m, as calculated in Section 3.2. The microstrip transmission feed line showed an impedance of 50 Ω for a standard SMA connector. After optimization, the size of the antenna, comprising a feed line and balun, was 43.5 mm × 42.5 mm. Figure 13 shows a photo, as well as the VSWR and gain of the antenna measured using a network analyzer. The impedance bandwidth of the wound-dressing antenna was 3.2–4.6 GHz when the return loss was less than −10 dB. The wound-dressing antenna exhibited a peak gain of 0.67 dBi at 3.8 GHz, and two transmission zeros at 2.5 and 5.7 GHz were found near the passband from the filtering balun. The 3 dB gain bandwidth was 3.2–4.6 GHz, which corresponded to the impedance bandwidth. The radiation efficiency was defined as the ratio of gain and directivity of the antenna, where directivity was calculated using the equation 32,400/(Θ_1d_Θ_2d_), where Θ_1d_ and Θ_2d_ are the half-power beam widths of the theta and phi directions, respectively, measured in degrees. Total efficiency of an antenna is defined as radiation efficiency multiplied by impedance mismatch [24]. The calculated radiation efficiency and total efficiency at 3.8 GHz were 47% and 42%, respectively. Gain can be increased by an antenna array [10] or by increasing the number of directors [16]. For biomedical applications, antennae are integrated with a transmitter or as a passive RFID tag. The power required for signal transmission can be provided by the reader or the TX chip. Figure 14 shows the far-field radiation pattern in the *yx*-plane and *zx*-plane at 3.8 GHz measured inside a far-field anechoic chamber with a rotary table. A stable radiation pattern with less backside radiation was observed. Notably, the quasi-Yagi antenna on the woun -dressing matched the simulation across the operating frequency bandwidth. Satisfactory similarities were obtained between the measurement and simulation results.

The wound-dressing-based antenna was flexible enough to provide coverage all over the body and conform to a cylinder examination of the bending effect. Two bending directions were examined: parallel and perpendicular to the dipole. Figure 15 shows the VSWR and gain of the antenna for flat, parallel, and perpendicular bending with bending fixtures with curvature radii of r = 5 and 6 cm. For parallel bending, the gain decreased from 0.67 to −1.22 dBi, and VSWR increased from 1.08 to 1.8 for a flat to 5 cm fixture, respectively. The gain decreased, and VSWR increased with the radius, as the director was bent toward the ground plane, resulting in a low gain, and even the dipole for a large radius resulted in lower gain at high frequency. Far-field radiation patterns in the *yx*- and the *zx*-planes at 3.8 GHz for parallel bending are shown in Figure 16. The directivity of the radiation pattern in parallel bending was almost identical, indicating that parallel bending only affected the dipole and director, resulting in lower gain. For perpendicular bending, the maximal gain decreased from 0.33 to −0.44 dB for flat to r = 5 cm fixtures. A total of 0.02 and 0.3 GHz frequency shifts at the lower band for r = 6 and 5 cm fixtures were observed after perpendicular bending. A slight frequency shift was observed as the resonance path of the director and dipole was changed undr perpendicular bending, resulting in a frequency shift. Lower gain was also observed in association with perpendicular bending with a poor resonance, which was smaller than that parallel bending. The far-field radiation patterns in the *yx*- and the *zx*-planes at 3.8 GHz for perpendicular bending are shown in Figure 17. An almost identical direction of radiation pattern was also observed under perpendicular bending. Therefore, stable radiation patterns were observed for the two bending directions of the wound-dressing-based antenna, indicating that the proposed flexible antenna was easily and efficiently fabricated and designed for wireless medical monitoring.

## 4. Conclusions

In this study, an inkjet-printed, wound-dressing-based quasi-Yagi antenna was successfully realized. The morphologies and electrical properties of the inkjet-printed silver metal film on a wound-dressing with a screen-printed UV-curable paste were optimized for high-frequency applications. The relative permittivity and loss tangent of the wound dressing were calculated as 3.15–3.25 and 0.04–0.05, respectively. The frequency band of the wound dressing antenna was 3.2–4.6 GHz, with a maximal gain of −2.8–0.67 dBi. An almost identical direction of the radiation pattern was observed for the two before bending, although gain and bandwidth were reduced after bending. This study demonstrates a wound-dressing antenna for medical remote monitoring that enables simple fabrication, disposable environmental protection, and low cost for mass production.

## Figures and Tables

**Figure 1 micromachines-13-01510-f001:**
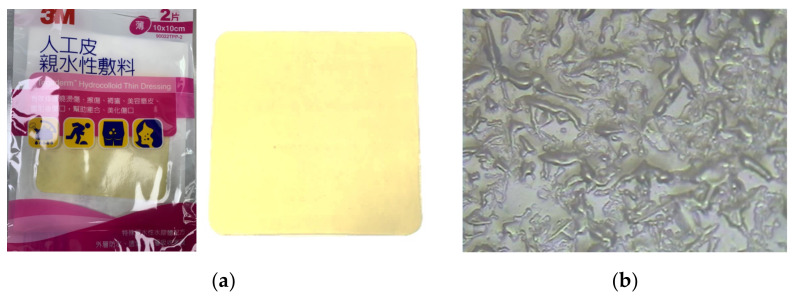
(**a**) Optical photographs and (**b**) surface morphology of hydrocolloid thin wound dressing.

**Figure 2 micromachines-13-01510-f002:**
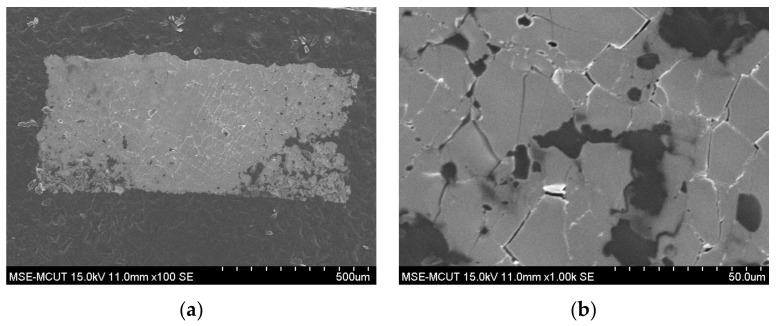
Surface morphology SEM image of inkjet-printed silver film on the wound dressing: (**a**) 100× and (**b**) 1000×.

**Figure 3 micromachines-13-01510-f003:**
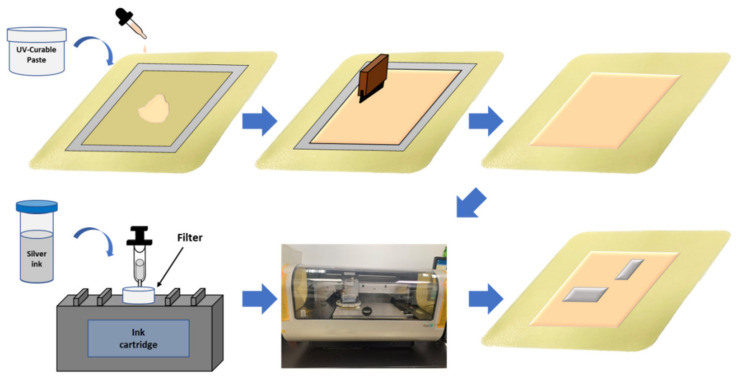
Schematic diagram of the inkjet printing process of silver film onto a hydrocolloid thin wound dressing.

**Figure 4 micromachines-13-01510-f004:**
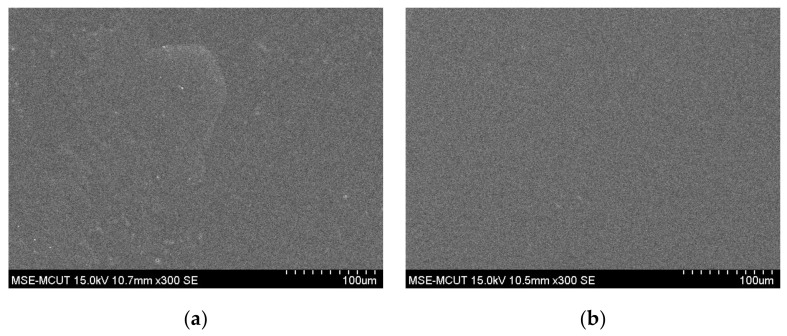
Surface morphology SEM images of screen-printed UV-curable paste on the wound-dressing: (**a**) one pass and (**b**) two passes.

**Figure 5 micromachines-13-01510-f005:**
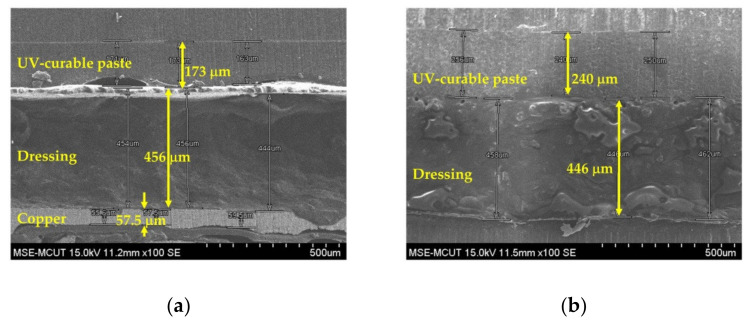
Cross section of SEM images for screen-printed UV-curable paste on the wound-dressing: (**a**) one pass and (**b**) two passes.

**Figure 6 micromachines-13-01510-f006:**
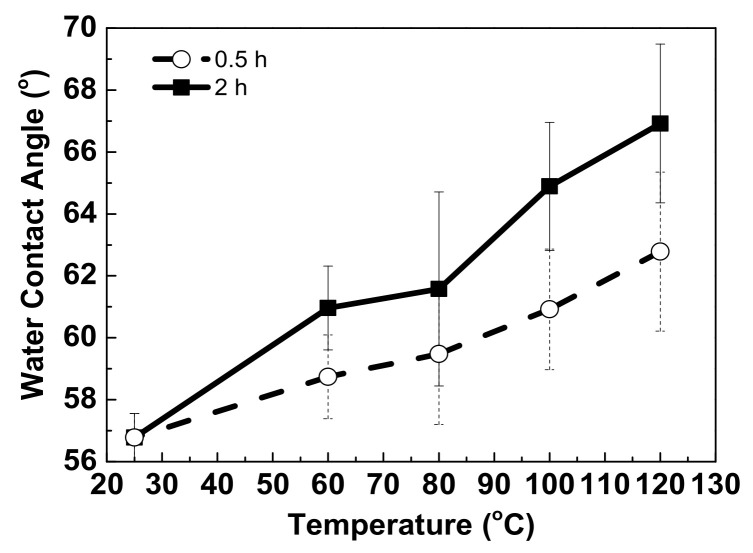
WCA versus temperature of hard baking after 2 h on UV-curable paste.

**Figure 7 micromachines-13-01510-f007:**
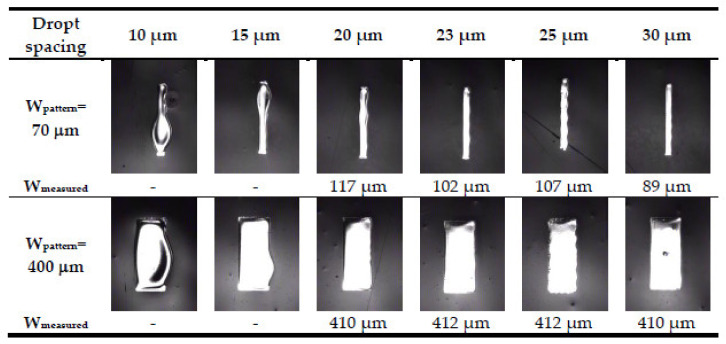
Optical photographs of silver films obtained with one-pass printing and a droplet spacing of 10–30 μm.

**Figure 8 micromachines-13-01510-f008:**
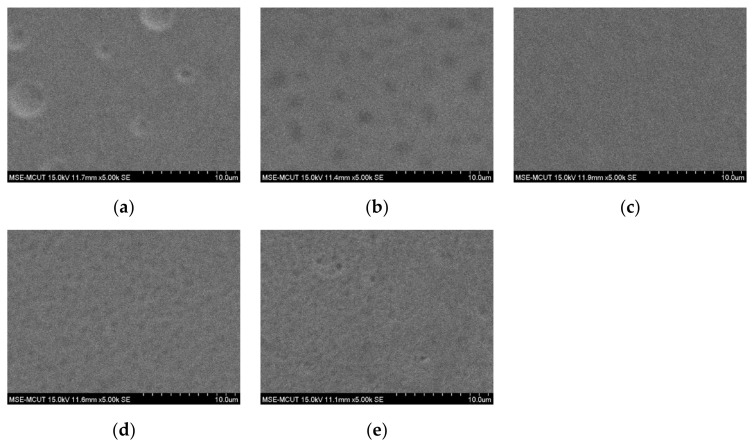
Surface morphologies of silver films obtained with (**a**) one, (**b**) two, (**c**) three, (**d**) four, and (**e**) five passes.

**Figure 9 micromachines-13-01510-f009:**
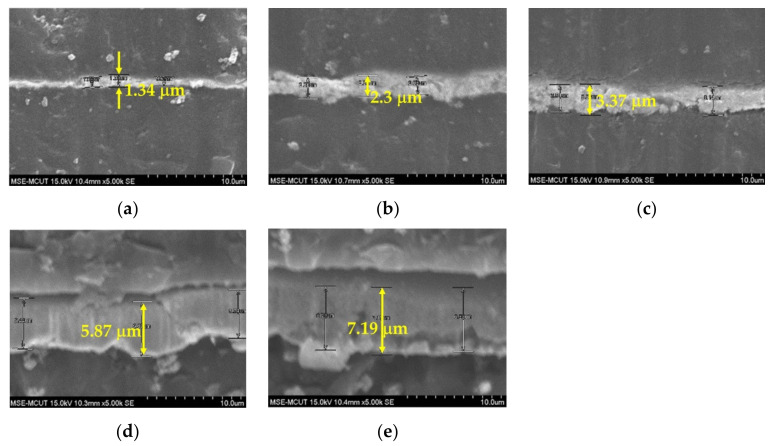
Cross-section views of silver films obtained with (**a**)one, (**b**) two, (**c**) three, (**d**) four, and (**e**) five passes.

**Figure 10 micromachines-13-01510-f010:**
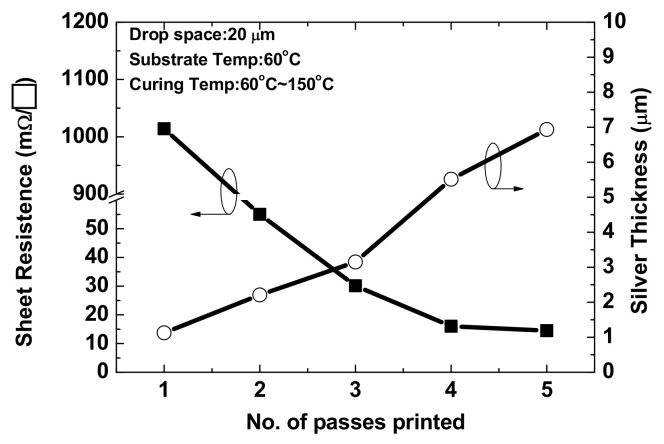
Sheet resistance and conductivity of silver films obtained with one to five-pass printing.

**Figure 11 micromachines-13-01510-f011:**
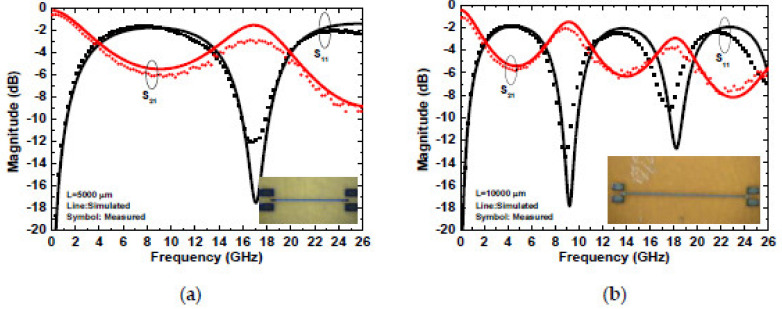
Simulated and measured S parameters of inkjet-printed silver lines on a wound dressing with lengths of (**a**) 5000 μm and (**b**) 10,000 μm.

**Figure 12 micromachines-13-01510-f012:**
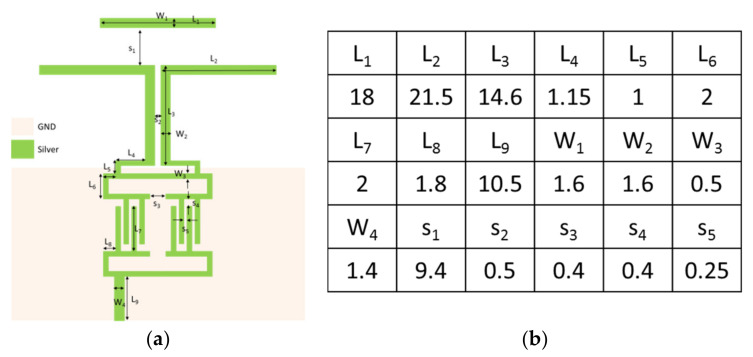
(**a**) Geometrical configuration of the proposed wound-dressing-based quasi-Yagi antenna. (**b**) Parameters of the designed antenna (mm).

**Figure 13 micromachines-13-01510-f013:**
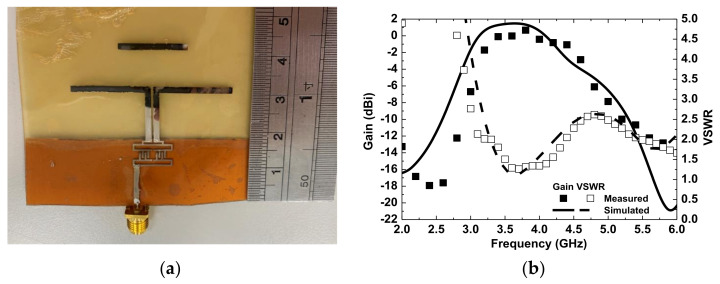
(**a**) Photo and (**b**) simulated and measured input VSWR and gain of the filtering antenna.

**Figure 14 micromachines-13-01510-f014:**
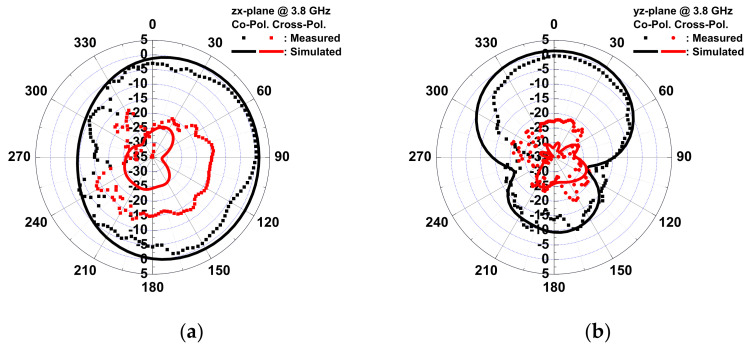
Simulated and measured radiation patterns of the proposed antenna in (**a**) the zx-plane and (**b**) the yz-plane at 3.8 GHz.

**Figure 15 micromachines-13-01510-f015:**
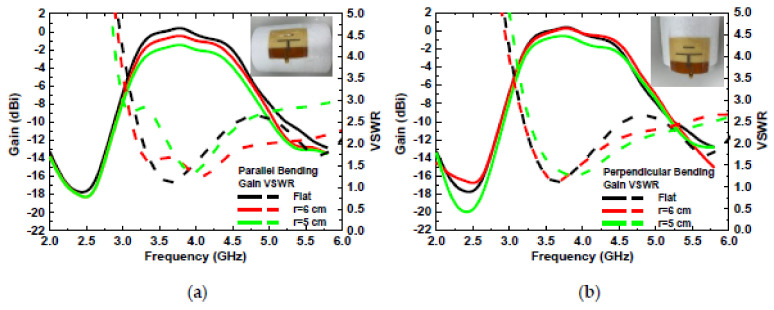
Measured input VSWR and gain of the antenna for (**a**) parallel and (**b**) perpendicular bending with dipole.

**Figure 16 micromachines-13-01510-f016:**
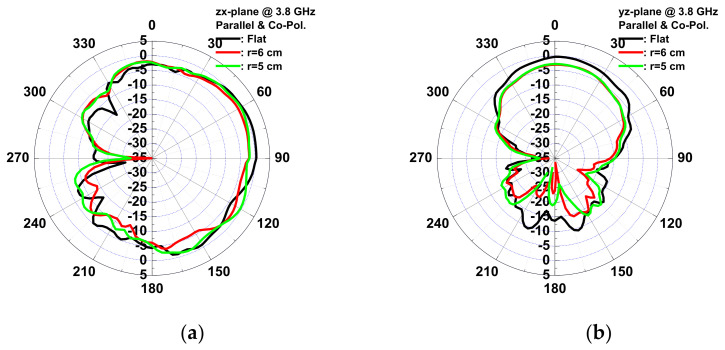
Measured copolarization radiation patterns of the antenna for parallel bending with dipole in (**a**) the zx-plane and (**b**) the yz-plane at 3.8 GHz.

**Figure 17 micromachines-13-01510-f017:**
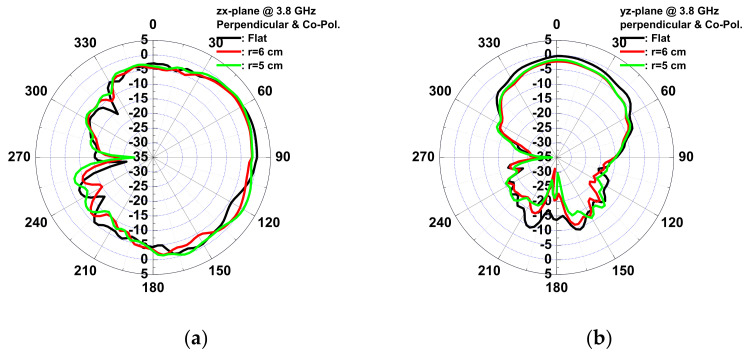
Measured copolarization radiation patterns of the antenna for perpendicular bending with dipole in (**a**) the zx-plane and (**b**) the yz-plane at 3.8 GHz.

**Table 1 micromachines-13-01510-t001:** Comparison of flexible antennae: this work and other published designs.

Reference	Frequency	Material Parameters			Process	Structure	Gain
	(GHz)	Substrate	e_r/_tanδ	Thickness			(dBi)
[4]	1.8/2.45	jean fabric + foam	1.7/0.085	2 mm	Stitching	Patch + EBG	-
[5]	1–3	Kevlar fabric + foam	2.6/0.006	25 mm	Weaving	Spiral	6.5
[6]	0.915	Cat. #A1220 + foam	1.26/0.007	4 mm	Stitching	Slotted patch	−3.56
[7]	5	PET/PES	2.2/0.01	0.36 mm	Embroidered–woven	SIW	−4.9
[8]	2.45	Polycotton	3.23/0.06	970 μm	Screen printing	Rectenna	-
[9]	2–5	Kapton HN	3.4/0.0018	76 μm	Screen printing	Quasi-dipole	2.3
[10]	77	PREPERM PPE260	2.6/0.0044	0.15 mm	Screen printing	Parasitic antenna array	11.2
[14]	60	PEN	2.9/0.025	125 μm	Inkjet printing	CPW patch	1.86
[15]	13	PDMS	2.68/0.04	180 μm	Inkjet printing	Patch	-
[16]	24.5	LCP	2.9/0.0025	100 μm	Inkjet printing	5-director Yagi–Uda	8
[17]	2.45	PET	2.7/0.135	125 μm	Inkjet printing	CPW-fed Z-shaped	1.44
This Work	3.2–4.6	Wound dressing	3.2/0.05	700 μm	Inkjet printing	Quasi-Yagi	0.67

## Data Availability

The data presented in this study are available in article.

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
