# Peer review of "Wound-Dressing-Based Antenna Inkjet-Printed Using Nanosilver Ink for Wireless Medical Monitoring"

_micromachines, 2022, doi:10.3390/mi13091510_

Round 1

Reviewer 1 Report

Paper: Wound-Dressing-Based Antenna Inkjet-Printed Using 2 Nano-Silver Ink for Wireless Medical Monitoring

Comments:

1. Potential applications of the antenna should be added in the abstract. The word wireless medical monitoring is too broad.

2. More quantitative analysis should be added to the abstract.

3. Why gain of the antenna is only 0.67 dBi? Is it possible to increase the gain?

4. What are radiation efficiency and total efficiency?

5. Simulation environment of the antenna is not clear. Please elaborate it.

6. Texts in Fig. 5 are invisible. Readability should be enhanced. 

Author Response

Dear Reviewer,

The valuable comments and suggestions from Reviewers have been incorporated in the revised manuscript. We would like to thank the reviewers sincerely for the time and effort spent to help improve the presentation of this paper. To strength this paper, we have modified and added the information in the revised manuscript (marked with shown highlighted in color). The point-by-point responses to the reviewers’ comments are in the attached file.

Best Regards,

Authors

Reviewer 2 Report

Inkjet printed antenna using nano-silver ink has been proposed. The presented work has certain merit and contributes to some extent to wireless medical monitoring. However, the paper needs to be further improved based on the following suggestions:

1- The introduction section needs more organization, it is not clear to the reader. Also, you can compare the listed antennas in this section and the proposed antenna in Table. 

2- in page 8, "Substituting the calculated relative permittivity and loss-tangent into the software, the measured and simulated results were found similar " What is the name of the software, and more details about the calculation method be appreciated?

3- Although the substrate has low tangential losses of 0.05, the gain of the antenna is very low. What about radiation efficiency? Is this sufficient gain at that low frequency for biomedical application? 

4- Why did the author select the yagi-uda antenna to implement it?

Author Response

(The authors gave the same response as above.)

Round 2

Reviewer 2 Report

No comments